# Efficacy of the New Inotropic Agent Istaroxime in Acute Heart Failure

**DOI:** 10.3390/jcm11247503

**Published:** 2022-12-18

**Authors:** Imma Forzano, Pasquale Mone, Gaetano Mottola, Urna Kansakar, Luigi Salemme, Antonio De Luca, Tullio Tesorio, Fahimeh Varzideh, Gaetano Santulli

**Affiliations:** 1Division of Cardiology, Department of Advanced Biomedical Sciences, “*Federico II*” University, 80131 Naples, Italy; 2Division of Cardiology, Department of Medicine, Wilf Family Cardiovascular Research Institute, Einstein Institute for Aging Research, Albert Einstein College of Medicine, New York, NY 10461, USA; 3Casa di Cura “*Montevergine*”, Mercogliano, 83013 Avellino, Italy; 4Department of Mental and Physical Health and Preventive Medicine, University of Campania “*Vanvitelli*”, 81100 Caserta, Italy; 5Department of Molecular Pharmacology, Einstein-Mount Sinai Diabetes Research Center (*ES-DRC*), Einstein Institute for Neuroimmunology and Inflammation (*INI*), Fleischer Institute for Diabetes and Metabolism (*FIDAM*), Albert Einstein College of Medicine, New York, NY 10461, USA

**Keywords:** acute heart failure, calcium, inotropic agents, istaroxime, lusitropic agents, Na^+^/K^+^-ATPase, NKA, PST2744, SERCA2a

## Abstract

Current therapeutic strategies for acute heart failure (AHF) are based on traditional inotropic agents that are often associated with untoward effects; therefore, finding new effective approaches with a safer profile is dramatically needed. Istaroxime is a novel compound, chemically unrelated to cardiac glycosides, that is currently being studied for the treatment of AHF. Its effects are essentially related to its inotropic and lusitropic positive properties exerted through a dual mechanism of action: activation of the sarcoplasmic reticulum Ca^2+^ ATPase isoform 2a (SERCA2a) and inhibition of the Na^+^/K^+^-ATPase (NKA) activity. The advantages of istaroxime over the available inotropic agents include its lower arrhythmogenic action combined with its capability of increasing systolic blood pressure without augmenting heart rate. However, it has a limited half-life (1 hour) and is associated with adverse effects including pain at the injection site and gastrointestinal issues. Herein, we describe the main mechanism of action of istaroxime and we present a systematic overview of both clinical and preclinical trials testing this drug, underlining the latest insights regarding its adoption in clinical practice for AHF.

## 1. Introduction

Heart failure (HF) is a clinical syndrome characterized by several symptoms, including dyspnea, ankle swelling, and fatigue, and signs, including peripheral edema, elevated jugular venous pressure, and pulmonary crackles. It is mainly due to structural and/or functional cardiac abnormalities causing a failure of the heart to pump enough to satisfy the metabolic requirements of the organism or generating an elevated intracardiac pressure to provide them [1].

With the aging of the population, HF incidence and prevalence are increasing worldwide [2,3]. HF incidence is about 6.0–7.9 per 1000 person-years in people >45 years old and about 21 per 1000 in people >65 years old [4,5]. The total percentage of people affected by HF is expected to rise from 2.4% in 2012 to 3.0% in 2030. HF prevalence is extremely variable with the lowest numbers in sub-Saharan Africa, but prevalence is projected to rise even in low- and middle-income countries as populations age and the burden of HF risk factors, such as elevated blood pressure, continues to increase [6,7]. HF has different classifications, based on the ejection fraction (EF), according to European Society of Cardiology (ESC) and the American Heart Association (AHA) guidelines. The ESC guidelines define HF as follows: HF with preserved ejection fraction (HFpEF) is characterized by an EF ≥ 50%, HF with mildly reduced ejection fraction (HFmrEF) is characterized by an 40% ≤EF < 49%, and HF with reduced ejection fraction (HFrEF) is characterized by an EF < 40%. The AHA defines: HFpEF with EF ≥ 50%; HFmrEF with 41% ≤ EF ≤ 49%; HF with improved EF (HFimpEF) for patients who have EF improved from a lower level to an EF >40% at follow-up; and HFrEF has EF ≤ 40% [1,8,9].

Acute HF (AHF) is universally considered as a gradual or rapid onset of signs and symptoms of HF resulting in a need for urgent therapy or hospitalization [8,10,11,12,13]. AHF can lead to cardiogenic shock, characterized by a life-threatening low-cardiac-output state causing end-organ hypoperfusion and hypoxia [8,14,15,16,17,18]. AHF in-hospital mortality ranges from 4 to 13% and the mortality rate at 1-year post-discharge is 25–30%, with more than 45% of readmissions [19,20,21,22,23]. 

Current therapy for AHF is based on inotropic agents that are often associated with serious adverse effects. Indeed, catecholamines, inhibitors of Na^+^/K^+^-ATPase (NKA), phosphodiesterase-3 inhibitors, and calcium (Ca^2+^) sensitizers, have been associated with tachycardia, ischemia, hypotension, and even with an excess of mortality, presumably related to arrhythmias in the short-term and the activation of signaling pathways that aggravate maladaptive remodeling of the failing heart in the long-term [24,25,26]. Therefore, there is an urgent need to identify new effective inotropic modulators with a safer profile. 

Istaroxime is a relatively novel compound, a derivative of androstenedione, is chemically unrelated to cardiac glycosides, and possesses inotropic and lusitropic actions exerted through a dual mechanism: inhibition of the Na^+^/K^+^-ATPase (NKA) and activation of the sarcoplasmic reticulum calcium ATPase isoform 2a (SERCA2a) [27]. 

## 2. NKA: Na^+^/K^+^-ATPase Pump

NKA, first described in 1957 [28], is a ubiquitous enzyme that actively transports three Na^+^-ions through the cell membrane outside the cytoplasm in exchange for two K^+^ ions imported inside the cytoplasm [29,30]. NKA is a pump made up of two subunits: α and β. Subunit β does not seem to contain functional sites, but it is necessary to stabilize the α subunit and to guarantee the passage from the endoplasmic reticulum to the cell membrane [30]. The transport is accomplished by enzyme conformational changes between two states, E1 and E2. Using energy derived from ATP hydrolysis, NKA produces electrochemical gradients across the membrane necessary for electrical excitability as well as cellular uptake of ions, nutrients, and neurotransmitters. Electrochemical gradients across the membrane are necessary to regulate cell volume as well as intracellular pH [31]. 

NKA is a target of cardiotonic glycosides (CG), such as ouabain, digoxin, and digitoxin, which can bind NKA in the E2 state, inhibiting it [32,33,34]. This inhibition causes an increase in intracellular [Na^+^]. Consequently, the Na^+^/Ca^2+^ exchanger (NCX) ejects Na^+^ in exchange for Ca^2+^, thereby increasing the cytosolic concentration of Ca^2+^, activating excitation-contraction coupling, ultimately enhancing myocardial contractility [24,32,35,36]. 

## 3. SERCA2a: Sarcoendoplasmic Reticular Adenosine Triphosphate-Driven Ca^2+^ Pump

The family of the sarco/endoplasmic reticulum Ca^2+^ ATPase gene, also known as SERCA, counts different isoforms. The SERCA2 gene is known to codify for SERCA2a-d, whereas SERCA2a encodes for a 997-aminoacid protein, a specific isoform located in the cardiac muscle, slow-twitch skeletal muscle, and smooth muscle cells [37]. In cardiomyocytes, SERCA2a is able to generate an influx of Ca^2+^ from the cytosol into the sarcoplasmic reticulum (SR) against the gradient. 

This process guarantees the relaxation of cardiac muscle fibers and a sufficient Ca^2+^ storage in the SR that can be utilized to start a new contractile activity for the ensuing contraction. 

SERCA2a is inhibited by phospholamban (PLB) and this regulation depends on its state of phosphorylation mediated by protein kinase A (PKA): in its de-phosphorylated form, PLB inhibits SERCA2a, whereas in its phosphorylated form SERCA2a is released from such inhibition [38,39]. Stimulation of β-adrenergic receptors causes the phosphorylation of PLB by PKA, inactivating PLB and stimulating SERCA2a, improving myocardial contraction and relaxation (inotropic and lusitropic positive effects [40,41]). These processes explain why the dysregulation of SERCA2a and PLB interaction is intimately related to HF, along with the desensitization and downregulation of myocardial β-adrenergic receptors due to their chronic activation [42,43,44,45,46,47,48,49,50,51]. 

## 4. Ca^2+^ and SERCA2a Function in Cardiac Contractility

The contractile activity of the heart is due to the interaction of myosin and actin filaments, an interaction finely tuned by cytoplasmic Ca^2+^ levels [24]. Ca^2+^ fluxes in the myocardium are dysregulated in HF, producing a depression of myocardial contractility. 

Ca^2+^ enters the cardiomyocyte during depolarization through L-type Ca^2+^ channels localized in the cell membrane in proximity of the SR. This Ca^2+^ influx induces a further release of Ca^2+^ from the SR in the cytoplasm, through intracellular Ca^2+^ release channels known as type 2 ryanodine receptors (RyR2) [52,53,54]. Thus, cytosolic [Ca^2+^] rises up to a critical concentration that activates the contractile system of the myocyte. To complete the contraction and start the relaxation phase, Ca^2+^ previously released by the SR needs to be re-uptaken. This step is possible thanks to SERCA2a, which brings intracytoplasmic Ca^2+^ into the SR against a concentration gradient [55,56,57]. 

Dysregulation of Ca^2+^ fluxes could be caused by alterations of RyR2 and/or loss of function of SERCA2a. Altered RyR2 generates an inappropriate diastolic release of Ca^2+^ from the SR, known as a Ca^2+^ leak; this diastolic leak reduces Ca^2+^ content in the SR and consequently, Ca^2+^ is available for the subsequent myocardial contraction [58,59,60,61]. 

Even the loss of function of SERCA2a reduces the availability of Ca^2+^ for the next contraction because of the inability of this pump to reuptake a sufficient amount of the ion. Moreover, SERCA2a loss of function compromises ventricular relaxation and causes diastolic dysfunction, reducing the quantity and speed of Ca^2+^ re-uptake from the cytosol [58,62].

## 5. Istaroxime

Istaroxime *-(E,Z)-3-[(2-aminoethoxy)imino] androstane-6,17-dione hydrochloride*- is a compound derived from androstenedione, hence considered a cardiotonic steroid, developed for the treatment of AHF [63]. Its application is limited to acute intravenous therapy due to its short plasma half-life (less than 1 h) because of the extensive hepatic processing that generates a long-lasting metabolite named PST3093 (E,Z)-[(6-beta-hydroxy-17-oxoandrostan-3-ylidene)amino]oxyacetic acid [64]. Istaroxime is a first-in-class drug with both inotropic and lusitropic effects without vasodilator properties; on the contrary, one of its characteristics is to rise systolic blood pressure (SBP) [65]. As mentioned above, istaroxime properties are related to its dual mechanism of action: NKA inhibition and SERCA2a stimulation. Istaroxime inhibits NKA by binding its E2 state from the extracellular side [66] (Figure 1).

Its inhibition of NKA is similar to that of digoxin [67] and leads to an increased [Ca^2+^], obtaining an inotropic effect. At nanomolar concentrations, istaroxime stimulates SERCA2a activity and Ca^2+^ uptake through a direct interaction with the SERCA2a/PLB complex, independent of cAMP/PKA and PLB phosphorylation [68] (Figure 2).

Henceforth, an influx of Ca^2+^ into the SR is generated during the diastolic phase, promoting myocardial relaxation, with a lusitropic effect [48,69,70] without inducing spontaneous Ca^2+^ efflux from the SR [68,71]. The increase in Ca^2+^ reserves through SERCA2a into the SR contributes to augment Ca^2+^ ions available for the following cardiac cycle, contributing to the inotropic effect of the compound [58]. The peculiar combination of istaroxime targets (Figure 3) seems to confer a better safety profile compared to digoxin; for instance, treatment with istaroxime has been associated with a lower risk of arrhythmias [72].

### Other Therapeutical Applications of Istaroxime

Istaroxime has also been shown to have anti-cancer actions due to its antiproliferative capacity exhibited in tumor cell lines [73]. Its use could be taken into account in prostate cancer [74]. Indeed, the use of androgen deprivation therapy in prostate cancer can be limited by the onset of cardiovascular events, including HF. In this sense, istaroxime could combine its antineoplastic and inotropic actions [75]; remarkably, the antiproliferative capacity of istaroxime is shared with other cardiotonic steroids, so these compounds are being proposed as new antitumoral drugs [34,76].

## 6. Preclinical Studies

In an animal model of diabetic cardiomyopathy, istaroxime was shown to improve diastolic dysfunction (DD) stimulating SERCA2a and reducing alterations in intracellular Ca^2+^ handling [77]; equally important, in animal models of acute decompensated HF it significantly improved hemodynamic and echocardiographic parameters [78]. Istaroxime was also compared to dobutamine in a preclinical study investigating chronic ischemic HF, showing to be an effective inotropic agent without positive chronotropic actions [79]. The chronic use of istaroxime was tested in a hamster model of progressive HF and was demonstrated to improve cardiac function and heart rate variability [80]. 

Comparing the electrophysiological effects of istaroxime and digoxin in guinea pig ventricular myocytes, istaroxime was shown to inhibit (−43%) the transient inward current (I_TI_) induced by the transient flux of Ca^2+^ in the presence of a complete block of the NKA pump; this effect was not evident with digoxin. Therefore, the therapeutic index of istaroxime may be accounted for by inhibition of I_TI_, a current directly involved in digitalis-induced arrhythmias [81,82,83,84]. Furthermore, the toxicity of istaroxime was compared to ouabain in murine cells, showing that istaroxime can reach a significant inotropic effect without activating Ca^2+^/calmodulin-dependent kinase II (CaMKII) whose over-stimulation could lead to cardiomyocyte death. Interestingly, at its inotropic concentration, istaroxime does not evoke a significant increase in diastolic local (Ca^2+^ sparks) or propagated (Ca^2+^ waves) SR Ca^2+^ release. In fact, istaroxime breaks arrhythmogenic Ca^2+^ waves into mini waves, which are less arrhythmogenic. This aspect corroborates the substantial safety compared to digitalis compounds. Remarkably, these results were evident even in PLB-knockout myocytes, suggesting that the safety of istaroxime is not merely related to its capacity to dissociate the SERCA/PLB interaction [85]. The main preclinical studies are summarized in Table 1.

## 7. Clinical Investigations

The first evaluation of istaroxime in humans was a phase I-II dose escalating study evaluating the safety and tolerability of istaroxime and its specific effects on ECG and hemodynamic parameters in patients with chronic HF with reduced systolic function [86]. Three cohorts of six patients were exposed to four sequentially increasing (0.005–5.0 μg/kg/min) 1-h infusions. In addition to safety, hemodynamic parameters were evaluated by an impedance cardiography, a digital Holter recorder, and by electrocardiography. Enhanced contractility was demonstrated with evidence of improvement of the acceleration index. Istaroxime improved the left cardiac work index, cardiac index, and pulse pressure at doses of ≥1 μg/kg/min, with evidence of activity at doses of 0.5 μg/kg/min. Istaroxime also shortened the QTc. The compound was tolerated at doses of up to 3.33 μg/kg/min with evidence of gastrointestinal symptoms and injection site pain at higher doses [86].

The HORIZON-HF (Hemodynamic, Echocardiographic, and Neurohormonal Effects of Istaroxime, a Novel Intravenous Inotropic and Lusitropic Agent) trial is a randomized, double-blind, placebo-controlled, dose-escalation trial (*NCT00616161*) in which patients were randomized to istaroxime or placebo (ratio 3:1) [87]; patients treated with istaroxime were randomized to 0.5 µg/kg/min, 1.0 µg/kg/min, or 1.5 µg/kg/min doses [88]. The primary endpoint was the change in pulmonary capillary wedge pressure (PCWP) compared with placebo after a 6 h continuous infusion [87]. All the three doses of istaroxime lowered PCWP (mean ± SD: −3.2 ± 6.8 mm Hg, −3.3 ± 5.5 mm Hg, and −4.7 ± 5.9 mm Hg compared to 0.0 ± 3.6 mm Hg with placebo; *p* < 0.05 for all doses); furthermore, secondary end points (changes in cardiac index, right atrial pressure, SBP, diastolic blood pressure (DBP), heart rate (HR), and stroke work index) improved. In addition, changes in left ventricular (LV) end-diastolic and systolic volumes, LV ejection fraction (EF), diastolic function indexes, neurohormones, renal function, troponin, pharmacokinetics, and safety were evaluated. A significant decrease in HR (*p* = 0.008, 0.02, and 0.006, with 0.5, 1.0, and 1.5 μg/kg/min, respectively) and an increased SBP (*p* = 0.005 and *p* < 0.001, with 1 and 1.5 μg/kg/min, respectively) were observed in patients treated with istaroxime. The cardiac index increased during the 1.5 μg/kg/min infusion vs placebo (*p* = 0.04), but not at the end of the 6 h infusion. At the ultrasound examination (echocardiography), the LV end-systolic volume was reduced in the 1.0 μg/kg/min istaroxime group compared to placebo (−15.8 ± 22.7 mL vs. −2.1 ± 25.5 mL; *p* = 0.03), and LV end-diastolic volume was reduced in the 1.5 μg/kg/min group compared to placebo (−14.1 ± 26.3 mL vs. +3.9 ± 32.4 mL; *p* = 0.02). E-wave deceleration time increased in the 1.5 μg/kg/min group (+30 ± 51 ms vs. +3 ± 51 ms; *p* = 0.04). There were no changes in neurohormones, renal function, or troponin I. Adverse events were dose-related gastrointestinal symptoms and injection site pain. In particular, vomiting and nausea occurred and were dose-related. No deaths occurred during the treatment period [89].

A recent phase II, multicenter, randomized, double-blind, placebo-controlled, parallel group investigation trial (*NCT02617446*) revealed that a 24 h infusion of istaroxime at 0.5 or 1.0 μg/kg/min in patients with AHF and reduced LVEF markedly improves LV diastolic and systolic function without major cardiac adverse effects [90]. The primary endpoint was the E/e’ ratio change from baseline to 24 h that was significantly reduced by both doses of istaroxime compared to placebo (cohort 1: −4.55 ± 4.75 istaroxime 0.5 μg/kg/min vs. −1.55 ± 4.11 placebo, *p* = 0.029; cohort 2: −3.16 ± 2.59 istaroxime 1.0 μg/kg/min vs. −1.08 ± 2.72 placebo, *p* = 0.009). Moreover, other parameters including E/A ratio, left atrial dimensions, and inferior cava diameter improved. Among others, there was a decrease in HR (from baseline by about 3 bpm with istaroxime 0.5 μg/kg/min and by 8–9 bpm with istaroxime 1.0 μg/kg/min with significant changes vs. placebo at 3 to 24 h in the high-dose group; 3 h: −10.61 ± 10.04, *p* < 0.001; 6 h: −8.89 ± 9.83, *p* = 0.001; 12 h: −9.49 ± 11.96, *p* = 0.005; 24 h: −9.61 ± 12.10, *p* = 0.004) and a significant increase in SBP with the 1.0 μg/kg/min dose (from baseline by about 3 mmHg with istaroxime 0.5 μg/kg/min and by 6–8 mmHg with istaroxime 1.0 μg/kg/min reaching statistical significance compared to placebo at 3 to 12 h in istaroxime 1.0 μg/kg/min group; 3 h: 7.63 ± 9.22, *p* < 0.001; 6 h: 8.08 ± 11.06, *p* = 0.001; 12 h: 9.00 ± 11.75, *p* = 0.006). These results occurred early after starting the infusion, with a significant difference already evident at 3h, most likely due to the pharmacokinetic profile of istaroxime, characterized by a rapid onset of action and a rapid washout after infusion termination. An increased estimated glomerular filtration rate (eGFR) was observed in the group treated with the high dose of the drug compared to placebo. Self-reported dyspnea and N-terminal pro-brain natriuretic peptide improved in all groups without significant differences between istaroxime and placebo [90].

Also in this trial, the most common adverse events were injection site reactions and gastrointestinal events, the latter primarily with istaroxime 1.0 μg/kg/min. A higher rate of patients experiencing abdominal pain, nausea, and vomiting was observed with istaroxime 1.0 μg/kg/min. At this dose, a high rate of injection site reaction was observed in patients with short intravenous catheters, leading to a necessary change of the peripheral line or use of peripheral long line or a central line. One case of treatment discontinuation due to injection site problems occurred. Neither major cardiovascular events nor an increase in arrhythmias occurred. These observations confirm the findings of the HORIZON-HF trial and further extend the safety of istaroxime to 24 h compared to the HORIZON-HF trial, in which the compound was tested only for 6 h [88,90]. 

The clinical effects of istaroxime appear to be favorable for patients with AHF with low or borderline SBP, for which inotropes have been associated with major cardiovascular adverse events and possibly an increased risk of mortality [91,92]. The safety and efficacy of istaroxime in patients with AHF-related pre-cardiogenic shock (stage B of the Classification of the Society for Cardiovascular Angiography and Interventions, SCAI) was tested in the SEISMiC trial (*NCT04325035*), a phase II, multicenter, randomized, double-blind, placebo-controlled, parallel group study [93]. The SEISMiC study was designed to compare the safety and efficacy of istaroxime with placebo in patients hospitalized for AHF-related SCAI stage B pre-cardiogenic shock with persistent hypotension (75 < SBP < 90 mmHg) but no clinical signs of hypoperfusion (both clinically and as evidenced by venous lactate levels <2.0 mml/L). Patients were randomized to continuous 24 h infusion of istaroxime 1.0 µg/kg/min or 1.5 µg/kg/min or placebo. The primary endpoint was the adjusted area under the curve (AUC) representing the change in SBP from baseline, start of study drug infusion, through 6 h. Secondary endpoints included: SBP AUC through 24 h, changes from baseline in SBP, in DBP, and mean arterial pressure (MAP) at 6 and 24 h; changes from baseline in HR, treatment failure score (treatment failure defined as death or need for circulatory, respiratory, or renal mechanical support or need for intravenous inotrope or vasopressor treatment); increase from baseline in SBP ≥ 5% and/or ≥ 10 mmHg; changes in quality of life (measured by the EuroQol 5 Dimension 5 Level, EQ-5D-5L), change from baseline to 24 h in echocardiography parameters. Secondary endpoints were also considered: changes in troponin and N-terminal pro-B-type natriuretic peptide (NT-proBNP), hospital readmission for HF and for any cause by day 30, in-hospital worsening HF to day 5, and length of in-hospital stay. Endpoints of safety were: incidence of adverse events, changes in vital signs, change in 12-lead ECG, incidence of supraventricular and ventricular arrhythmias, changes in laboratory parameters, renal function, cardiac troponin (I and T), and mortality through day 30. In the SEISMiC trial, istaroxime increased SBP and improved echocardiographic measures, including an increase in the cardiac index and a reduction in the left atrial and left ventricular dimensions. The adjusted mean 6 h AUC was 53.1 (standard error [SE] 6.88) mmHg × hour in the istaroxime group vs. 30.9 (SE 6.76) mmHg × hour in the placebo group (*p* = 0.017); an increase of 72% was observed. The adjusted mean 24 h SBP AUC was 291.2 (SE 27.5) mmHg × hour in istaroxime arm *vs* 208.7 (SE 27.0) mmHg × hour in placebo arm (*p* = 0.025), with an increase of 40%. The adjusted SBP increase at 6 h was 12.3 (SE 1.71) mmHg in the istaroxime group vs. 7.5 (SE 1.64) mmHg in the placebo group (*p* = 0.045). The corresponding adjusted changes in SBP at 24 h were 17.1 (SE 2.36) mmHg and 15.1 (SE 2.25) mmHg in the istaroxime group vs. placebo (*p* = 0.543). Increases were noted in DBP and MAP, as well. Of note, the concomitant increase in both the cardiac index (at 24 h: +0.16 ± 0.1 vs. −0.06 ± 0.1 L/min/m^2^; *p* = 0.016) and SBP had not been observed with any previous intravenous drugs administered to patients with cardiogenic shock related to AHF. Additionally, other echocardiographic measures besides the cardiac index that demonstrated improvements at 24 h in the istaroxime group as compared to the placebo group were: left atrial area (−1.8 ± 0.5 vs. 0.0±0.5 cm^2^; *p* = 0.008), LV end-systolic volume (−8.7 ± 4.2 vs. 3.3 ± 4.2 mL; *p* = 0.034), and LV end-diastolic volume (−6.5 ± 4.9 vs. 5.6 ± 4.8 mL; *p* = 0.061). Laboratory parameters did not suggest an effect mediated by istaroxime on end-organ damage. Istaroxime treatment was associated with more adverse events (such as nausea, vomiting, and pain at infusion site) than placebo, but it was not associated with arrhythmias or worsening of renal function. Among gastrointestinal adverse events, nausea was the most frequent (28%), followed by vomiting (14%). Injection site pain occurred in 14% of the patients [93].

In order to reduce gastrointestinal adverse effects and injection site pain related to istaroxime, several attempts have been made, including the development of a liposomal formulation of the molecule, encapsulating istaroxime in a drug delivery system conveniently designed to be quickly destabilized in plasma in order to minimize alterations of the pharmacokinetic profile of istaroxime. *Poly ethylene glycol 660-hydroxystearate* (PEG-HS) was chosen as an excipient to modulate the bilayer fluidity and the release properties of the liposomes, obtaining an almost complete release in physiological conditions in less than 10 min [94]. It is important to emphasize that istaroxime use may also be considered in pediatric patients [95].

In summary, istaroxime was shown to be a potential new inotropic agent, safer than currently available treatments for AHF. Its ability to improve overall cardiac function in HF with reduced arrhythmogenic risk launched a new field of investigation in AHF treatment, which has most recently led to the development of other molecules with highly selective SERCA2a activation and longer half-time starting from istaroxime long-lasting metabolite PST3093 [64,96]. The main clinical trials investigating istaroxime are summarized in Table 2.

Intriguingly, therapies based on cardiac contractility modulation (CCM) take advantage of SERCA2a upregulation in patients with chronic HF. CCM therapy is based on an implantable device that delivers a non-excitatory high-voltage bipolar signal to the right ventricle (RV) synchronized on the absolute refractory period of the action potential [97,98]. It stimulates SERCA and RyR upregulation, PLB phosphorylation, and downregulation of NCX. Its modulation of Ca^2+^ flux and increase of re-uptake of the ion into SR results in the increase of myocardial contractility [97]. CCM has been approved by the FDA based on the results of several randomized clinical trials [99,100,101,102,103,104], which revealed that CCM improves the signs and symptoms of HF, particularly in patients with a LVEF between 25% and 45%, NYHA III symptoms despite guideline-directed medical therapy, and a sinus rhythm with normal QRS length. CCM therapy is associated with a reduction in hospitalization for HF compared with the rate of hospitalization the year before the device implantation [105,106]. Furthermore, a recent study revealed that CCM improves LVEF, global longitudinal strain, and myocardial mechano-energetic efficiency in patients with HFrEF [107]. These pieces of evidence suggest that the production of a metabolite derived from istaroxime, harboring a selective SERCA2a action and a longer half-time, could overcame the pharmacodynamic limitations of istaroxime itself, making it a new effective pharmacologic tool not only for AHF but for chronic HF as well [108].

## 8. Future Perspectives

HF is a clinical syndrome with considerable medical implications in our society, both in terms of morbidity and mortality [109,110,111]. Specifically, de novo AHF and acute decompensated HF (ADHF) represent life-threatening conditions [112,113,114,115]. According to recent evidence examining its effectiveness and safety, it is reasonable to consider istaroxime as the first example of a new useful and safe category of drugs against HF in contraposition with traditional inotropic agents whose uses are often limited by several adverse effects. The development of new molecules derived from istaroxime with longer lifetime and less adverse effects is ongoing and current results are quite promising. Nonetheless, more investigations are warranted.

## 9. Conclusions

Istaroxime is a cardiotonic steroid currently under investigation for AHF treatment. Compared to the inotropic agents presently used for AHF in clinical practice with which it shares inotropic actions, it also has a lusitropic positive effect and a better safety profile. Its properties are exerted through a dual mechanism of action: activation of SERCA2a and inhibition of NKA activity. Istaroxime is being tested exclusively for acute intravenous therapy due to its half-time of only 1 h because of its rapid hepatic metabolism. Available data indicate that istaroxime has an overall safe profile with a reduced arrhythmogenic risk. Adverse events include gastrointestinal discomfort, most likely attributable to the systemic inhibition of NKA, and pain at the injection site. Hitherto, only studies on animal models and phase I and II trials are available; therefore, albeit there are promising perspectives for istaroxime as a new inotropic agent for AHF treatment, it is necessary to further expand the investigation on this compound to assess its effectiveness and long-term safety.

## Figures and Tables

**Figure 1 jcm-11-07503-f001:**
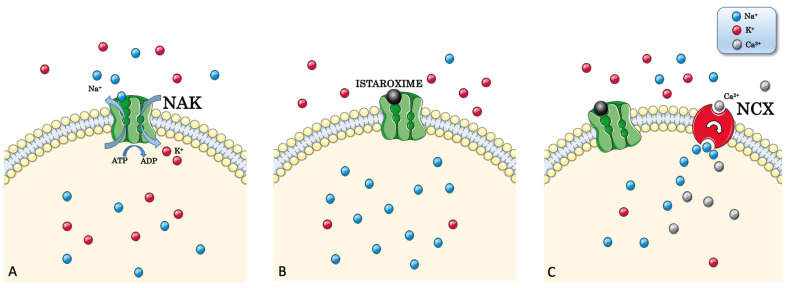
**Istaroxime inhibits the Na^+^/K^+^-ATPase (NAK) pump.** (**A**) NAK actively transports three Na^+^-ions through the cell membrane outside the cytosol in exchange for two K^+^ ions inside the cytoplasm. (**B**) Istaroxime inhibits NAK by binding it from the extracellular side with a consequent raise of [Na^+^] in the intracellular side. (**C**) Increased [Na^+^] leads to the activation of the Na^+^/Ca^2+^ exchanger (NCX), which exchanges three Na^+^ ions for one Ca^2+^ ion, eventually increasing intracytoplasmatic [Ca^2+^].

**Figure 2 jcm-11-07503-f002:**
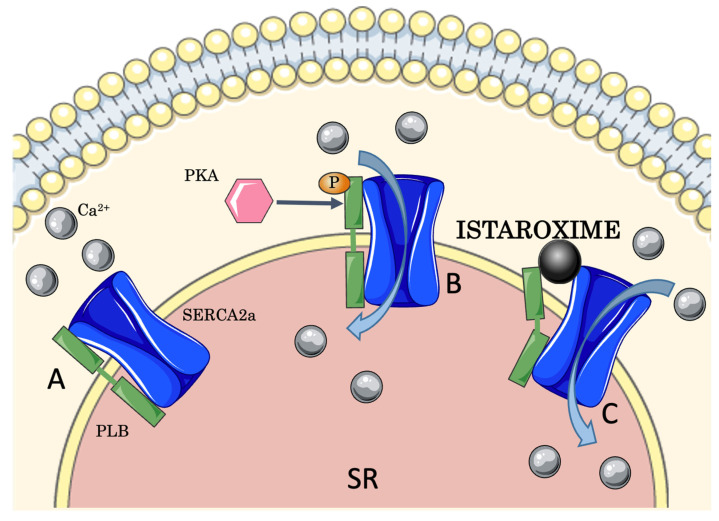
**Istaroxime activates SERCA2a.** (**A**) PLB, in its dephosphorylated form, inhibits SERCA2a. (**B**) PKA phosphorylates PLB releasing SERCA2a inhibition, thereby Ca^2+^ can be re-uptaken from the cytosol to the sarcoplasmic reticulum (SR). (**C**) Istaroxime activates SERCA2a with a direct interaction with the SERCA2a/PLB complex, independent of PLB phosphorylation. PKA: protein kinase A; PLB: phospholamban; SERCA2a: sarco/endoplasmic reticulum Ca^2+^ ATPase 2a.

**Figure 3 jcm-11-07503-f003:**
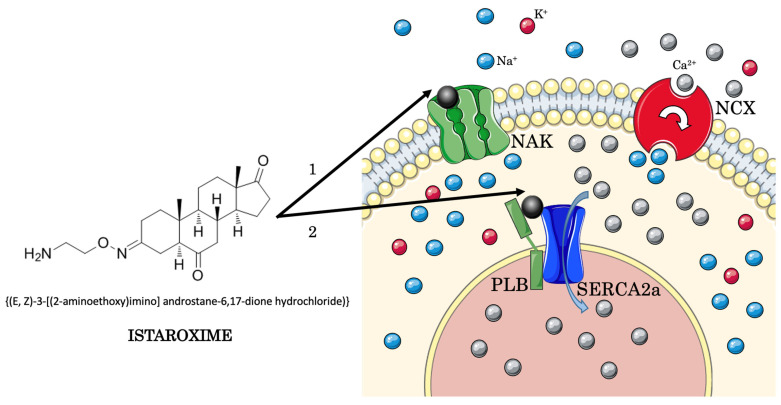
**Dual mechanism of action of istaroxime.** Istaroxime (structure depicted on the left) inhibits NAK by binding it from the extracellular side. NAK inhibitions increase [Na^+^] inside the cytoplasm, activating NCX that expels three Na^+^ -ions in one Ca^2+^ ion exchange, raising [Ca^2+^] inside the cytoplasm, increasing myocardial contractility (inotropic effect); istaroxime activates SERCA with a direct interaction with the complex formed by SERCA2a and PLB, independent of PLB phosphorylation. SERCA2a stimulation triggers an influx of Ca^2+^ against the gradient from the cytosol to the SR, guaranteeing the relaxation of cardiac muscle fibers (lusitropic effect) and a sufficient amount of Ca^2+^ storage in the SR that can be utilized to start a new contractile activity for the ensuing contraction (empowerment of inotropic effect). NAK: Na^+^/K^+^-ATPase pump; NCX: Na^+^/Ca^2+^ exchanger; PKA: protein kinase A; PLB: phospholamban; SERCA: sarco/endoplasmic reticulum Ca^2+^ ATPase.

**Table 1 jcm-11-07503-t001:** Summary of designs and conclusions of preclinical studies investigating istaroxime.

Clinical Condition	Animal Model	Endpoint	Conclusion
**DCM**	DCM STZ induced in rats	DD	Istaroxime improved DD stimulating SERCA2a and reducing alterations in intracellular Ca^2+^ handling.
**ADHF**	Canine model of HF produced by multiple sequential intracoronary embolizations with microspheres	LVEF (%); LVEDV (mL); LVESV (mL).	Istaroxime improved hemodynamic and echocardiographic parameters.
**Chronic ischemic HF, comparing istaroxime to dobutamine**	Canine model of HF produced by ligation of the left anterior descending coronary artery and intracoronary embolizations	LV function	Istaroxime was shown to be an effective inotropic agent without positive chronotropic actions.
**Progressive HF**	Hamster model of progressive HF	Heart/body weight ratio; max d*P*/d*T*; min d*P*/d*T*; LVSP; CFR	Istaroxime improved cardiac function and heart rate variability
**Electrophysiological effects of istaroxime and digoxin**	Guinea pig isolated ventricular myocytes	Effects on I_TI_	Istaroxime inhibited I_TI_ (effect not evident with digoxin)
**Cardiotoxic effects of equi-inotropic concentrations of istaroxime and ouabain**	Rat isolated ventricular myocytes	Cell viability; Apoptosis; CaMKII activation.	Istaroxime had a significant inotropic effect, neither activating CaMKII nor promoting cardiomyocytes death (contrary to digoxin)

ADHF: acute decompensated heart failure; CaMKII: Ca^2+^/calmodulin-dependent kinase II; CFR: coronary flow rate; DCM: diabetic cardiomyopathy; DD: diastolic dysfunction; d*P*/d*T*: derivative of LV pressure; LVEDV: LV end-diastolic volume; LVESV: LV end-systolic volume; HF: heart failure; I_TI_: transient inward current of Ca^2+^; LV: left ventricle; LVEF: LV ejection fraction; LVSP: LV systolic pressure; MI: myocardial infarction; STZ: streptozotocin.

**Table 2 jcm-11-07503-t002:** Results of the main clinical trials on Istaroxime.

Clinical Trial	Primary Endpoint	Main Results
**HORIZON-HF (*NCT00616161*)**	Change in PCWP (mmHg)	Istaroxime: −3.2 ± 6.8, −3.3 ± 5.5, and −4.7 ± 5.9 vs. placebo: 0.0 ± 3.6; *p* < 0.05 (for all doses)
**The Clinical Study of the Safety and Efficacy of Istaroxime in Treatment of ADHF (*NCT02617446*)**	E/e’ ratio change from baseline to 24 h	cohort 1: istaroxime 0.5 μg/kg/min: −4.55 ± 4.75 vs. placebo: −1.55 ± 4.11, *p* = 0.029; cohort 2: 59 istaroxime 1.0 μg/kg/min: −3.16 ± 2. vs. placebo: −1.08 ± 2.72, *p* = 0.009
**SEISMiC (*NCT04325035*)**	AUC (mmHg × hour; change in SBP from baseline through 6 h)	Istaroxime: 53.1 ± 6.88 vs. placebo: 30.9 ± 6.76, *p* = 0.017

ADHF: acute decompensated heart failure; AUC: area under curve; PCWP: pulmonary capillary wedge pressure; SBP: systolic blood pressure.

## Data Availability

Not applicable.

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
