# Peer review of "Efficacy of the New Inotropic Agent Istaroxime in Acute Heart Failure"

_jcm, 2022, doi:10.3390/jcm11247503_

Round 1
Reviewer 1 Report
Dear authors/editor,
I have read with great interest the review article entitled "Efficacy of the New Inotropic Agent Istaroxime in Acute Heart 2 Failure" by Imma Forzano et al. submitted for publication in Journal of Clinical Medicine. Overall, it is a comprehensive, well-written article, which presents thoroughly a novel inotropic agent, istaroxime, and its therapeutic implications. Nevertheless, several issues should be addressed prior to any other evaluation.
1. The authors write Istaroxime instead of istaroxime, namely with capital as first letter in every single use. They should avoid it and use it properly.
2. The authors state that the main adverse effects of istaroxime in clinical studies are gastrointestinal effects and injection pain. Although they are not serious, authors should present them more comprehensively.
3. Are there any cardiac diseases, in which istaroxime should not be administered?
4. HF is frequently complicated with atrial fibrillation. Are there any implications in such a setting?
5. A "Future perspective" section should be added. 6. It would be useful if the authors could summarize in one paragraph the non CV therapeutic implications of istaroxime.
In conclusion, the authors should be congratulated for their review on a very interesting topic. If they could improve the previous points, then their article could be considered for acceptance.
Author Response
REVIEWER 1
I have read with great interest the review article entitled "Efficacy of the New Inotropic Agent Istaroxime in Acute Heart 2 Failure" by Imma Forzano et al. submitted for publication in Journal of Clinical Medicine. Overall, it is a comprehensive, well-written article, which presents thoroughly a novel inotropic agent, istaroxime, and its therapeutic implications. Nevertheless, several issues should be addressed prior to any other evaluation.
R: Thank you for your words of appreciation.
- The authors write Istaroxime instead of istaroxime, namely with capital as first letter in every single use. They should avoid it and use it properly.
R: We thank the Reviewer for her/his comment. In the revised version of the paper we used “istaroxime” instead of “Istaroxime”.
- The authors state that the main adverse effects of istaroxime in clinical studies are gastrointestinal effects and injection pain. Although they are not serious, authors should present them more comprehensively. R: Thank you for the insightful piece of advice. We added more detailed information about adverse effects.
- Are there any cardiac diseases, in which istaroxime should not be administered? R: We thank the Reviewer for the question. In the current literature, cardiac diseases in which istaroxime should not be administered are not clearly described yet.
- HF is frequently complicated with atrial fibrillation. Are there any implications in such a setting? R: Thank you for the interesting question; In the clinical trials that we have analyzed, implications in patients affected by, or complicated with, atrial fibrillation are not described.
- A "Future perspective" section should be added. R: Thank you for the suggestion. We will add a “Future perspective” section.
- It would be useful if the authors could summarize in one paragraph the non CV therapeutic implications of istaroxime. R: Thanks. We added a paragraph about non CV therapeutic implication of istaroxime.
In conclusion, the authors should be congratulated for their review on a very interesting topic. If they could improve the previous points, then their article could be considered for acceptance. R: Thank you very much.
Reviewer 2 Report
jcm-2085904, Efficacy of the New Inotropic Agent Istaroxime in Acute Heart Failure by Forzano et al. The authors aimed to review and present the current literature of clinical trials testing Istaroxime, underlining the latest insights regarding its adoption in clinical practice for treatment of acute heart failure. This is an interesting study and the authors have collected a unique dataset supporting the aim of their work. The review paper is generally well written and structured.
Comments:
Abstract:
Page 1, line 27: “..we present the current literature of clinical trials…” the reviewer suggests adding “..we present the current literature of preclinical and clinical trials…” since the authors already discussed the preclinical studies in their review as well.
Introduction:
- Page 1, line 41: The reviewer suggests replacing the phrase “HF is going to rise...” with “HF is expected to rise...”
- Page 1, line 45: For clarity, the abbreviations ESC and AHA should be fully explained at their first mention in the manuscript.
- Page 2, line 52, The reviewer suggests adding more details about “AHF” like its prevalence, mortality rate, …etc.
- Page 2, line 63: “Istaroxime is a relatively novel compound, derivative of…… calcium ATPase isoform 2a (SERCA2a)”. The reviewer suggests moving this paragraph under the “Istaroxime” title on page 3. The reviewer also suggests moving the titles:
“2. NKA: Na+/K+-ATPase Pump”, “3. SERCA2a: Sarcoendoplasmic Reticular Adenosine Triphosphate-Driven Ca2+ Pump”, and “4. Ca2+ and SERCA2a Function in Cardiac Contractility” under the “Istaroxime” title on page 3 as well to help the flow of the manuscript.
- Figures 3 and 4: make sure to include all abbreviations in each figure legend.
- If the authors are not the originators of the figures, seek permission to reproduce figures which were already published and/ or add reference.
- Page 5, line 178: “Chronic Istaroxime”? could the authors clarify?
- Page 9, line 342: “These pieces of evidence suggest that development of a Istaroxime metabolite with selective SERCA2a action and longer half-time could be a new pharmacologic tool for chronic HFrEF either”. The sentence is not clear. What do the authors try to indicate by this?
Author Response
REVIEWER 2
jcm-2085904, Efficacy of the New Inotropic Agent Istaroxime in Acute Heart Failure by Forzano et al. The authors aimed to review and present the current literature of clinical trials testing Istaroxime, underlining the latest insights regarding its adoption in clinical practice for treatment of acute heart failure. This is an interesting study and the authors have collected a unique dataset supporting the aim of their work. The review paper is generally well written and structured.
R: Thank you very much for your favorable comments.
Comments:
Abstract:
Page 1, line 27: “..we present the current literature of clinical trials…” the reviewer suggests adding “..we present the current literature of preclinical and clinical trials…” since the authors already discussed the preclinical studies in their review as well. R: Thank you for the suggestion.
Introduction:
- Page 1, line 41: The reviewer suggests replacing the phrase “HF is going to rise...” with “HF is expected to rise...” R: We thank the Reviewer for the suggestion. We replaced it as recommended.
- Page 1, line 45: For clarity, the abbreviations ESC and AHA should be fully explained at their first mention in the manuscript. R: Thank you for the advice, we spelled them out.
- Page 2, line 52, The reviewer suggests adding more details about “AHF” like its prevalence, mortality rate, …etc. R: Thanks. We added more details about AHF in the revised version of the manuscript.
- Page 2, line 63: “Istaroxime is a relatively novel compound, derivative of…… calcium ATPase isoform 2a (SERCA2a)”. The reviewer suggests moving this paragraph under the “Istaroxime” title on page 3. The reviewer also suggests moving the titles:
“2. NKA: Na+/K+-ATPase Pump”, “3. SERCA2a: Sarcoendoplasmic Reticular Adenosine Triphosphate-Driven Ca2+ Pump”, and “4. Ca2+ and SERCA2a Function in Cardiac Contractility” under the “Istaroxime” title on page 3 as well to help the flow of the manuscript. R: Thank you for the suggestion. We thought about this structure of the manuscript to introduce molecular pathways involved in HF before describing istaroxime. This idea relies on letting the Reader to analyze istaroxime properties after having read the molecular bases of HF. We respectfully believe that that changing the structure of the manuscript could affect our intentions.
- Figures 3 and 4: make sure to include all abbreviations in each figure legend. R: Thank you for the note. Rectified.
- If the authors are not the originators of the figures, seek permission to reproduce figures which were already published and/ or add reference. R: Thank you. The figures are original.
- Page 5, line 178: “Chronic Istaroxime”? could the authors clarify? R: We meant “Chronic use of istaroxime; rectified.- Page 9, line 342: “These pieces of evidence suggest that development of a Istaroxime metabolite with selective SERCA2a action and longer half-time could be a new pharmacologic tool for chronic HFrEF either”. The sentence is not clear. What do the authors try to indicate by this? R: Thank you for asking. Istaroxime is a compound designed for acute heart failure only due to its short half-time and continuous intravenous administration. We wanted to indicate that the development of a drug with a longer half-time could overcame these limitations related to the pharmacodynamics of the compound making it useful for chronic heart failure, too. We clarified the sentence in the reviewed version of the manuscript.
Round 2
Reviewer 1 Report
I have read the revisioned version of the manuscript. The authors have incorporated my recommendations.
Kind regards,